# LED Light Quality of Continuous Light before Harvest Affects Growth and AsA Metabolism of Hydroponic Lettuce Grown under Increasing Doses of Nitrogen

**DOI:** 10.3390/plants10010176

**Published:** 2021-01-19

**Authors:** Yubin Zhang, Lingyan Zha, Wenke Liu, Chengbo Zhou, Mingjie Shao, Qichang Yang

**Affiliations:** 1Institute of Environment and Sustainable Development in Agriculture, Chinese Academy of Agricultural Sciences, Beijing 100081, China; 82101175084@caas.cn (Y.Z.); zhaly@sjtu.edu.cn (L.Z.); 82101181045@caas.cn (C.Z.); 82101182120@caas.cn (M.S.); yangqichang@caas.cn (Q.Y.); 2Key Lab of Energy Conservation and Waste Management of Agricultural Structures, Ministry of Agriculture and Rural Affairs, Beijing 100081, China; 3School of Agriculture and Biology, Shanghai Jiaotong University, Shanghai 200240, China; 4Institute of Urban Agriculture, Chinese Academy of Agriculture Science, Chengdu 610213, China

**Keywords:** ascorbic acid metabolism, nitrogen levels, light quality, continuous light, LEDs

## Abstract

To study the effects of light quality of continuous light before harvest on the growth and ascorbic acid (AsA) metabolism of lettuce (*Lactuca sativa* L.) grown under relative high nitrogen level, lettuce plants grown under different nitrogen levels (8, 10 and 12 mmol·L^−1^) were subjected to continuous light with different red: blue light ratios (2R:1B and 4R:1B) before harvest. The results showed that the shoot fresh weight of lettuce under 12 mmol·L^−1^ nitrogen level was significantly higher than that under other treatments. There were no significant differences in shoot dry weight, root fresh weight, root dry weight, soluble sugar content, nitrate content and AsA content in leaves among the treatments at different nitrogen levels. The content of AsA in leaves was significantly higher than that in petioles before and after continuous light. Under the same nitrogen level, the fresh weight of lettuce under continuous light quality 4R:1B was significantly higher than that under other treatments. The content of AsA in lettuce leaves increased in different degrees after continuous light before harvest. High yield and AsA content could be obtained by 72 h continuous light with red and blue light 4R:1B at 12 mmol·L^−1^ nitrogen level. After continuous light, the content of AsA increased significantly due to the increase of the ratio of red light and nitrogen level, which increased the activities of L-galactono-1,4-lactone dehydrogenase (GalLDH) and dehydroascorbic acid reductase (DHAR) involved in AsA synthesis and in the recycling of DHAR to AsA respectively.

## 1. Introduction

Nitrogen (N) is an element which is a component of proteins, nucleic acid, hormones and many other important substances in plants, which plays a significant role in plant metabolism [1]. The production of leafy vegetables requires high doses of nitrogen fertilizer, and higher yields could be obtained by applying high-concentration nitrogen fertilizer, especially in hydroponic cultivation mode. However, excessive nitrogen could alter the plant physiology primarily the photosynthetic capacity of plants [2]. The application of extensive N fertilizers, however, may lead to the consumption of a large amount of carbohydrates, resulting in the decrease of plant biomass under high nitrogen supply, and the growth and development of plants are limited [2]. This phenomenon is even more prominent under low light conditions, mainly manifested in low AsA content and excessive accumulation of nitrogen [3]. Previous studies have shown that the enhancement of the hydroponic leafy vegetables’ quality can be achieved with the employment of methods such as cutting off nitrogen before harvest, adding appropriate proportion of ammonium nitrogen in nutrient solution [4]. However, it is not conducive to the accumulation of dry matter and the faster increase of yield in hydroponic leafy vegetables. The production of hydroponic leafy vegetables rich in ascorbic acid (AsA) and low nitrate content is critical for the improvement of the cultivation systems towards the production of high-quality products for the consumers. Continuous light treatment before harvest can not only improve the quality of vegetables, but also increase the yield of vegetables. Some studies have shown that contradictive results exist on the effect of the levels of N fertilization on the AsA levels [5,6]. Moreover, higher nitrogen level than recommended doses of N can even result in the reduction of the levels of the antioxidant [7]. However, the dynamic responses of lettuce grown under high nitrogen levels to continuous light with different light qualities remain unclear. Sufficient nitrogen could promote the photosynthesis of plants, which is closely related to the light environment of plants [8].

Continuous light is a special light mode that can maximize the lighting time, providing plants with lighting for 24 h or over a period of more than 24 h in most cases [9,10]. A large number of studies have shown that continuous light can damage the photosynthesis many plant species [10,11]. A growing body of evidence, however, shows that continuous light could accelerate the growth, improve the yield and quality of specific crops [3]. This can be achieved by the improvement of the photosynthetic capacity via the prolonging the time of light exposure, strengthening the light intensity and by selecting the proportion of light quality that stimulates plant growth. Further-more, it could promote the rate of nitrogen assimilation and promote the growth of plants. Continuous light and nitrogen level are two powerful components. The study further strengthens the evidence that the continuous light before harvest could effectively regulate the nitrate content and improve the nutritional quality of lettuce [12]. Light quality makes a valuable contribution to plants morphogenesis, growth and development. The effect of LED light quality on AsA content and metabolism is more intricate because its spectrum can be combined in different proportions. Red light and blue light are the main spectral constituents, so there are many studies on the effect of them on AsA. Compared with fluorescent lamp treatment, red and blue light treatment could visibly increase the AsA content of lettuce [13]. The results showed that the maximum yield of lettuce was obtained at the nitrogen level of 6 mmol·L^−1^, and the concentration of ascorbic acid and nitrate reached the highest at 8 mmol·L^−1^ [14]. Continuous light could prolong the light cycle to the maximum extent, which is a fruitful method to enhance the amount of light in protected horticulture. There is a report that the content of AsA in lettuce increased by 82% after continuous light for 48 h [12]. 

Ascorbic acid (AsA) is also known as vitamin C, which widely exists in plant organs and is related to multiple physiological processes such as stress resistance, cell division and photosynthesis [15]. AsA also plays an essential role in human health and normal life activities. Ascorbic acid cannot be synthesized in humans and animals on account of the lack of L-gulono-γ-lactone oxidase [16,17]. The principal synthetic pathway of AsA is galactose pathway [18]. L-galactono-1,4-lactone dehydrogenase (GalLDH) is the relevant enzyme in the AsA synthesis pathway of plants, and its activity determines the level of AsA content. AsA in plants is mainly oxidized by ascorbate peroxidase (APX) and ascorbate oxidase (AO) to monodehydroascorbic acid (MDHA). Hence, two molecules of MDHA are oxidized/reduced spontaneously producing a molecule of AsA and a molecule of DHA, and Monodehydroascorbate reductase (MDHAR) is reducing enzymatically MDHA to AsA using NADH as a co-factor. The oxidized DHA can be reduced to AsA or hydrolyzed to 2, 3-diketogulonic acid by dehydroascorbic acid reductase (DHAR) [19]. Oxidized glutathione (GSSG) can be to reduced glutathione (GSH) via glutathione reductase (GR), thus maintaining the content of GSH in plants. GSH is needed to provide electron donor in the process of DHA reduction and regeneration to AsA [20]. The content of AsA in plants is the result of synthesis, degradation and transportation [21]. The anabolism of AsA is affected by a variety of environmental factors, for example, light and temperature have a significant effect on the accumulation of AsA, and the light environment has a particularly important effect on the AsA metabolic pathway [22,23]. Although people have realized the significance of light in regulating AsA metabolism in plants, the mechanism by which it regulates AsA remains to be explored.

Leafy vegetables are nitrogen-loving crops, and an effective way to increase their yield is to enhance the concentration of nitrate nitrogen in the cultivation conditions. It can exert the best coupling effect of the two on the yield and quality of hydroponic lettuce to coordinate the nitrogen level and light environment treatment, and then achieve high-quality and high-yield crops in the process of plants growth. The aim of this study was to explore the response of lettuce growth, ascorbic acid content and metabolism under high nitrogen and continuous light conditions. In order to expound the regulation of ascorbic acid metabolism by short-term continuous light before harvest, the effects of continuous light on synthesis, oxidation and regeneration were measured from its content and related enzyme activities. It is hoped that this study could provide a theoretical basis for the regulation of continuous light quality in the production of high-yield and high-quality vegetables in plant factories, and provide valuable view into the interaction between continuous light quality and nutrient solution with high nitrogen level on the content and metabolism of AsA in lettuce.

## 2. Materials and Methods

### 2.1. Plant Materials and Growth Condition

Lettuce plants (*Lactuca sativa* L. cv. ‘Yidali’) were cultivated in a plant factory at 25 ± 1/20 ± 1 °C day and night temperature, 60% ± 5% relative humidity. In the experiment, the seedlings were seeded in sponge block, and then transplanted into hydroponic pots (180 × 60 × 6 cm^3^) after the second true leaf grew out completely. In terms of planting density, there were about 36 lettuce plants per square meter. Then, the lettuce was exposed to red and blue LED lights. According to previous studies, the light quality was 4R:1B, which was suitable for lettuce growth [24]. The light board was placed 40 cm above the cultivation tank, which light intensity was 150 μmol·m^−2^·s^−1^, and the light-dark cycle was set to 16/8 h. The experiment set the nitrogen level treatments of 8, 10, 12 mmol/L (N8, N10 and N12). Continuous light can improve the yield and quality of lettuce, which has been mentioned in many studies. The main purpose of this study was that light quality of continuous light before harvest affects growth and AsA metabolism of hydroponic lettuce grown under increasing doses of nitrogen. Therefore, this study took the sampling of the starting point of continuous light as the comparison object. A total of 17 days after transplantation, eight of them were randomly sampled for determination of initial value and the others were used to grow under continues light. The samples before 72 h continuous light treatment were taken as the comparison object. The composition ratios of red light and blue light in the two light quality treatments were set to 2:1 (ACL (2R:1B)) and 4:1 (ACL (4R:1B)), respectively. The light intensity was 150 μmol·m^−2^·s^−1^. A red and blue LED panel (50 × 50 cm^2^, Shenzhen Huihao Optoelectronic Co., Ltd., Shenzhen, China) was adopted, and the peak wavelengths were 655 nm and 430 nm respectively. 150 red beads and 150 blue beads were evenly staggered on apiece LED panel. The light intensity was measured at the height of the vegetable canopy by a light sensor recorder (Li-1500; Li-COR Biosci., Lincoln, NE, USA). The basic components of the nutrient solution were as follows:4 mmol·L^−1^ Ca(NO_3_)_2_, 0.75 mmol·L^−1^ K_2_SO_4_, 0.5 mmol·L^−1^ KH_2_PO_4_, 0.1 mmol·L^−1^ KCl, 0.65 mmol·L^−1^ MgSO_4_·7H2O, 1.0 × 10^−3^ mmol·L^−1^ H_3_BO_3_, 1.0 × 10^−3^ mmol·L^−1^ MnSO_4_·H_2_O, 1.0 × 10^−4^ mmol·L^−1^ CuSO_4_·5H_2_O, 1.0 × 10^−3^ mmol·L^−1^ ZnSO_4_·7H_2_O, 0.1 mmol·L^−1^ EDTA-Fe and 5 × 10^−6^ mmol·L^−1^ (NH_4_)_6_Mo_7_O_2_4·4H_2_O (pH: 5.7; EC: 1.3,1.5,1.7 mS·cm^−1^). The different gradient of nitrogen level was determined by the content of KNO_3_, and then KCl was added to supplement K^+^ in nutrient solution.

### 2.2. Sampling and Growth Parameter Measurements

A total of eight lettuce plants were randomly selected from each treatment at 6:00 (beginning and end of continuous light) on the 18th and 21th days. Shoot fresh weight, shoot dry weight, root fresh weight and root dry weight of 4 plants were measured. The lettuces were sterilized at 100 °C for 20 min, dried at 80 °C to constant weight, and the dry weight of shoot and root were weighed respectively. The remaining 4 plants were reserved for physiological admeasurement. After the leaves and petioles of the remaining 4 plants were separated from each other, the leaves and petioles of the remaining 4 plants were quickly frozen with liquid nitrogen and grounded into powder at low temperature. The samples were stored in the refrigerator at −80 °C. 

### 2.3. Soluble Sugar, Nitrate and AsA Determination

The soluble sugar content was determined by the sulfuric acid-phenol method [25]. 0.1 g of fresh plant sample powder was taken, 1.5 mL of distilled water was added, and extracted in a boiling water bath. Then we drew 0.5 mL of the sample solution in a test tube, added 1.5 mL of distilled water, and added phenol, concentrated sulfuric acid solution, color developed, and we determined and calculated the content of soluble sugar.

The nitrate content was determined by the sulfuric acid-salicylic acid method [26]. 0.1 g of fresh plant samples were taken, 1.5 mL of distilled water was added, and extracted in a boiling water bath. The 0.1 mL extract was drawn into a 10 mL test tube, and 0.4 mL of 5% salicylic acid-concentrated sulfuric acid solution was added and mixed. Then, 9.5 mL of 8% NaOH solution was added, and the nitrate content was determined and calculated after cooling.

The AsA content was measured by UPLC (Waters Corp, Milford, MA, USA) with an Acquity UPLC^®^ HSS T3 (2.1 × 50 mm, 1.8 μm) column [27]. Extraction solution: 1.5% metaphosphoric acid (Sigma-Aldrich, St. Louis, MO, USA), 4% CH_3_COOH (Sinopharm Group, Shanghai, China), 0.5 mmol·L^−1^ ethylene diamine tetraacetic acid (Aladdin, USA); buffer, 200 mmol·L^−1^ Tris (hydroxymethyl) aminomethane (Sigma-Aldrich, St. Louis, MO, USA); reducing solution, 750 mmol·L^−1^ dithiothreitol (Solarbio, Beijing, China); reaction stop solution, 0.4 mmol·L^−1^ H_2_SO_4_ (Sinopharm Group, Shanghai, China). Mobile phase: 0.1% formic acid (Sinopharm Group, Shanghai, China) solution (prepared with ultrapure water); flow rate: 0.25 mL·min^−1^; injection volume:2 µL; column temperature 25 °C; sample room temperature: 10 °C; detection wavelength: 245 nm; running time:1.5 min. The experimental steps were as follows: fresh leaf samples (0.1 g) were homogenized by 1 mL cold extraction solution. The homogenate was extracted by ultrasonic ice bath for 20 min, then centrifuged for 10 min. Afterwards, 50 µL supernatant was collected and mixed with 190 µL Tris buffer (200 mmol·L^−1^). Then, 10 µL of reducing solution was added and incubated at 25 °C for 45 min. Finally, 50 µL of reaction stop solution was added and shake well. The mixture was filtered via 0.22 μm microporous filters (Jinteng Co. Ltd., Tianjin, China). 

### 2.4. Enzyme Extraction and Assay 

The activity of APX was determined according to Cao’s method [28]. As for the extracts of DHAR, MDHAR and GR, 1 mmol·L^−1^ ethylene diamine tetraacetic acid, 0.1 mmol·L^−1^ diamine tetraacetic acid, 0.1% triton × 100 (Sigma-Aldrich, St. Louis, MO, USA), 0.2% mercaptoethanol (Sigma-Aldrich, St. Louis, MO, USA) and 2% polyvinyl pyrrolidone (Sigma-Aldrich, St. Louis, MO, USA) should be added into 50 mmol·L^−1^ PBS (PH = 7.5) buffer (Sinopharm Group, Shanghai, China). 0.1 g lettuce sample was put into a 2 mL centrifuge tube, 1 mL of extraction solution was added, mixed, ultrasonic ice bath extraction for 20 min, centrifugation at 15,000 rpm for 20 min, and the supernatant was the extract with enzyme. DHAR reaction solution was 100 mmol·L^−1^ HEPES-KOH (Aladdin, USA) (pH = 7.0), mixed buffer containing 1 mmol·L^−1^ EDTA and 2.5 mmol·L^−1^ GSH (Solarbio, China), and 6 mmol·L^−1^ DHA (Sigma-Aldrich, St. Louis, MO, USA) solution. Determination method: we sequentially added 2.8 mL buffer, 0.1 mL DHA solution, and 0.1 mL extraction solution to the cuvette, immediately mixed, measured the kinetic rate at 265 nm, and counted the number of times every 20 s. The MDHAR reaction solution was 50 mmol·L^−1^ HEPES-KOH (pH = 7.6) mixed buffer containing 0.5 mmol·L^−1^ AsA, 3 mmol·L^−1^ Nicotinamide adenine dinucleotide (Sigma-Aldrich, St. Louis, MO, USA) solution and 5 U·ml^−1^ Ascorbic acid oxidase (Sigma-Aldrich, St. Louis, MO, USA) solution. The determination method was to sequentially add 2.7 mL buffer, 0.1 mL nicotinamide adenine dinucleotide solution, 0.1 mL ascorbic acid oxidase (Sigma-Aldrich, St. Louis, MO, USA) solution, and 0.1 mL extract to the cuvette, mix immediately, measure the kinetic rate at 340 nm, and record once every 20 s. The GR reaction solution was 100 mmol·L^−1^ Tris-HCL (pH = 8.0) mixed buffer solution containing 1 mmol·L^−1^ ethylene diamine tetraacetic acid, 30 mmol·L^−1^ GSSG (Solarbio, China) solution and 6 mmol·L^−1^ nicotinamide adenine dinucleotide phosphate (Sigma-Aldrich, St. Louis, MO, USA) solution. The determination method was to sequentially add 2.6 mL buffer, 0.1 mL GSSG solution, 0.1 mL nicotinamide adenine dinucleotide phosphate solution, and 0.2 mL extraction solution to the cuvette, vortex immediately to mix, and measure the kinetic rate at 340 nm every 20 s count the number of times [29]. GalLDH enzyme activity was measured with a kit (Solarbio, China).

### 2.5. Statistic Analysis 

Data was analyzed by the statistical analysis software SPSS 25.0 (LSD, α = 0.05), and was graphed and analyzed by Graph Pad Prism 6.0.

## 3. Results 

### 3.1. Biomass Index

As shown in Figure 1, the shoot fresh weight (SFW) and shoot dry weight (SDW) of lettuce increased slightly with the increase of nitrogen level, before continuous light. It was at 8 mmol·L^−1^ nitrogen level that the SFW and SDW of lettuce were the lowest, which were 13.39 g and 0.64 g, respectively. When the nitrogen level was 12 mmol·L^−1^, the fresh weight and dry weight of shoot were the largest, which were 16.87 g and 0.73 g, respectively. The treatment of nitrogen level had no significant effect on the root fresh weight (RFW) and root dry weight (RDW) before continuous light treatment. SFW, SDW, RFW and the RDW of lettuce was significantly increased by continuous light treatment compared with that before treatment. The yield index of ACL (4R:1B) was higher than that of ACL (2R:1B) under the same nitrogen level. The results showed that ACL (4R:1B) treatment and nitrogen 12 mmol·L^−1^ treatment had the greatest influence on the yield of lettuce. SFW, SDW, RFW and RDW of lettuce were the largest, reaching 28.93, 1.31, 5.39 and 0.28 g, under this combination treatment. ACL (2R:1B) treatment and nitrogen 8 mmol·L^−1^ treatment had the least effect on lettuce. SFW, SDW, RFW and RDW of lettuce were the lowest, which were 20.47 g, 1.19 g, 3.11 g and 0.18 g, respectively. Therefore, the higher the nutrient liquid nitrogen level, the greater the yield of lettuce, under the ACL (4R:1B) treatment.

The content of soluble sugar and nitrate in lettuce under different high nitrogen levels was significantly affected by LED continuous light quality before harvest. The nitrogen level had no significant effect on soluble sugar content of lettuce leaves and petioles before continuous light (Figure 2a). The content of soluble sugar in petiole was significantly affected by LED continuous light quality before harvest. The ACL (4R:1B) treatment and the nitrogen 10 mmol·L^−1^ treatment had the greatest impact on the soluble sugar content of lettuce leaves. The maximum soluble sugar content of lettuce leaves reaches 17.50 mg·g^−1^ under this combination treatment. The ACL (4R:1B) treatment and 12 mmol·L^−1^ nitrogen treatment had the greatest impact on the soluble sugar content of lettuce petioles. Under this combination treatment, the maximum soluble sugar content of lettuce petioles reached 14.35 mg·g^−1^. The nitrogen level had no significant influence on the nitrate content of lettuce leaves and petioles before continuous light (Figure 2b). The nitrate content of leaves and petioles was significantly affected by LED continuous light quality before harvest under different nitrogen levels. The ACL (4R:1B) treatment and the nitrogen 10 mmol·L^−1^ treatment had the greatest impact on the nitrate content of lettuce leaves after continuous light. The nitrate content of lettuce leaves was the lowest 237.78 mg·kg^−1^ under this combination treatment. The ACL (4R:1B) treatment and the nitrogen 12 mmol·L^−1^ treatment had the greatest impact on the nitrate content of lettuce petioles after continuous light. The nitrate content of lettuce leaves was the lowest 341.28 mg·kg^−1^ under this combination treatment. Therefore, continuous light before harvest can significantly increase the soluble sugar content of lettuce and significantly reduce the nitrate content of lettuce.

### 3.2. Ascorbate and Glutathione Pools

As shown in Figure 3, nitrogen level had no significant effect on AsA content in leaves and petioles of lettuce before continuous light. The content of AsA and DHA in lettuce was significantly affected by LED continuous light quality before harvest. The ACL (4R:1B) treatment and 12 mmol·L^−1^ nitrogen treatment had the greatest impact on the AsA content of lettuce leaves. The maximum AsA content of lettuce leaves reached 3.35 mg·g^−1^ under this combination treatment. The ACL (4R:1B) treatment and 10 mmol·L^−1^ nitrogen treatment had the greatest impact on the AsA content of lettuce petioles. The maximum AsA content of lettuce leaves reached 0.77 mg·g^−1^ under this combination treatment. The nitrogen level has no significant effect on the DHA content of lettuce leaves before continuous light. However, the DHA content of petioles under 8 mmol·L^−1^ nitrogen level was significantly higher than that under other nitrogen levels. The DHA content in leaves and petioles was significantly affected by LED continuous light quality before harvest under different nitrogen levels. Continuous light quality ACL (2R: 1B) treatment and nitrogen level 8 mmol·L^−1^ treatment had the greatest effect on the DHA content of lettuce leaves.

### 3.3. The Activity of GalLDH in The Synzyme of AsA

As shown in Figure 4, the nitrogen level had a significant effect on the activity of GalLDH in lettuce leaves before continuous light. The GalLDH enzyme activity reaches the highest 0.157 U mg^−1^ FW when the nitrogen level was 10 mmol·L^−1^. Nitrogen level has no significant effect on GalLDH enzyme activity in petioles. The activities of GalLDH in leaves and petioles were significantly affected by LED continuous light quality before harvest under different nitrogen levels. The ACL (4R:1B) treatment and 12 mmol·L^−1^ nitrogen treatment had the greatest effect on the activity of GalLDH in lettuce leaves. The GalLDH enzyme activity of lettuce leaves reached a maximum (0.318 U mg^−1^ FW) under this combination treatment. The ACL (4R:1B) treatment and 10 mmol·L^−1^ nitrogen treatment had the greatest effect on the GalLDH enzyme activity of lettuce petioles. The AsA content in the leaf stalk of lettuce reached a maximum (0.132 U mg^−1^ FW) under this combination treatment.

### 3.4. The Activity of Key Enzymes Involved in AsA Recycling

As shown in Figure 5, Figure 5a showed that nitrogen level had a significant effect on the activity of APX enzyme in lettuce leaves before continuous light, and the highest APX enzyme activity was 0.425 U mg^−1^ FW at 10 mmol·L^−1^. Nitrogen level had no significant effect on the activity of APX in lettuce petiole. The APX activity of leaves and petioles under different nitrogen levels was significantly affected by LED continuous light quality before harvest. The ACL (4R:1B) treatment and 10 mmol·L^−1^ nitrogen treatment had the greatest impact on the activity of APX enzyme. The activity of APX in lettuce leaves and petioles were 0.467 and 0.174 U mg^−1^ FW under this combination of treatment. Figure 5b showed that nitrogen level had a significant effect on the activity of MDHAR in leaves and petioles of lettuce before continuous light. At 10 mmol·L^−1^, the highest activity of MDHAR in leaves was 2.20 U mg^−1^ FW, and that in petioles was 0.52 U mg^−1^ FW at 12 mmol·L^−1^. The activities of MDHAR in leaves and petioles were significantly affected by LED continuous light quality before harvest under different nitrogen levels. The ACL (4R:1B) treatment and 8 mmol·L^−1^ nitrogen treatment had the greatest effect on the activity of MDHAR, as well as the combination of The ACL (2R:1B) treatment and 12 mmol·L^−1^ nitrogen treatment. The activity of MDHAR in lettuce leaves was 2.37 U mg^−1^ FW. The activity of MDHAR in petiole decreased after continuous light. Figure 5c shows petioles before continuous light. The DHAR enzyme activity of leaves and petioles was the highest, 1.51 and 1.37 U mg^−1^ FW at 8 mmol·L^−1^. The ACL (4R:1B) treatment and 10 mmol·L^−1^ nitrogen treatment had the greatest impact on the DHAR enzyme activity of lettuce from the perspective of the entire growth period. The DHAR enzyme activity of lettuce leaves under this combination was 1.77 U mg^−1^ FW. The activity of petiole DHAR enzyme decreased after continuous light. Figure 5d showed that nitrogen level had no significant effect on GR activity of lettuce before continuous light. However, the GR activity of leaves and petioles under different nitrogen levels was significantly affected by LED continuous light quality before harvest. Continuous light quality ACL (2R:1B) treatment and nitrogen level of 10 mmol·L^−1^ had the greatest effect on GR enzyme activity. The lowest GR enzyme activity of lettuce leaves under this combination was 0.30 U mg^−1^ FW. The GR enzyme activity of petiole increased after continuous light. The lowest value was 0.09 U mg^−1^ FW under the ACL (2R:1B) treatment and 8 mmol·L^−1^ nitrogen treatment.

## 4. Discussion 

Nitrogen is the most important structural material in plants. The increase in nitrogen content can stimulate plant growth and obtain higher dry matter content [30]. Nitrogen deficiency would obstruct the synthesis of protein and decrease the activity of enzyme affect the photosynthesis and other life activities of leaves, and eventually lead to the plant growth retardation [31]. However, excessive application of nitrogen will lead to root growth retardation, photosynthetic energy reduction and slow growth of vegetables [32,33]. The shoot fresh weight under 12 mmol·L^−1^ nitrogen level was significantly higher than that under other treatments before continuous light treatment. There were no significant differences in other yield indexes among different nitrogen levels. This showed that the biomass accumulation has reached the asymptote, and would not continue to increase with the increase of nitrogen level [8]. Therefore, the optimization of nitrogen assimilation of hydroponically cultivated plants is particularly important. 

Light is also one of the important factors affecting plant growth and quality [34]. Continuous light is a convenient and efficient means to control the light environment in plant factory. Although most studies have shown that continuous light can cause damage to plants, reasonable setting of the light environment parameters of continuous light could effectively promote the growth and quality of hydroponic leafy vegetables [35]. Continuous light could promote the yield of sweet pepper [36], the growth of young tea plants [37] and the germination rate of alfalfa seeds [9]. Therefore, it is particularly important to explore the effect of continuous light quality on the growth of leafy vegetables under different gradient nitrogen levels. The fresh weight and dry weight of lettuce increased significantly after continuous light before harvest. This is according with the former research results [3,38,39], which may be because the increase of photosynthetic duration to improve the net photosynthetic efficiency of lettuce leaves and promote the accumulation of photosynthetic products, thus significantly improving the yield of lettuce [9]. The fresh weight of lettuce was significantly increased after continuous light treatment with different light quality. However, the fresh weight of ACL (4R: 1B) was significantly higher than that of ACL (2R:1B) only at 12 mmol·L^−1^, nitrogen level. Zha et al. [35] found that continuous light with high ratio of red to blue light quality could promote the growth of lettuce, which was consistent with the results of this experiment. It is because red light is the most absorbed light by green plants [39]. Red light plays a role in regulating light morphogenesis through phytochrome, which can promote stem elongation, promote carbohydrate synthesis, make plants grow faster and increase plant yield [40]. The yield of lettuce increased with the increase of the ratio of red light to continuous light before harvest. At 8 and 10 mmol·L^−1^, There was the trend that the yield of ACL (4R: 1B) treatment was higher than that of ACL (2R: 1B) treatment. This may be due to the interaction between the nutrient level of liquid nitrogen and the light quality of continuous light on lettuce. Photosynthesis is the energy source for carbon and nitrogen metabolism in plants, so light can regulate carbon and nitrogen metabolism in plants through photosynthesis, thereby affecting the content of nitrogen-containing compounds. On the contrary, since nitrogen is a component of photosynthetic proteins and pigments, so the content and activity of photosynthetic proteins and pigments can be changed by altering the nitrogen level, which affects the photosynthesis and biomass of plants [41]. The interaction of light and nitrogen can not only improve the light utilization efficiency of plants, but also promote the absorption and utilization efficiency of nitrogen fertilizer by plants. 

Nitrate content is one of the important quality indexes of hydroponic leafy vegetables. Compared with that before continuous light, the nitrate content of lettuce decreased significantly after continuous light. It is may be that Continuous light promoted the activity of nitrate reductase activity in lettuce leaves, resulting in a significant decrease in nitrate content [42]. On the other hand, this may explain that continuous light increases carbohydrate synthesis and increases the accumulation of ferredoxin and NADPH, which could reduce the content of nitrate in lettuce [38]. The nitrate content in the petiole of lettuce under the same treatment was significantly higher than that in the leaf. The difference of water content in different organs of plants is one of the important factors that lead to the difference of nitrate content in different organs. Because the water content in petiole is higher than that in leaf, the nitrate content in petiole is higher than that in leaf [43]. In addition, *NRT1.4*, as a low affinity transporter for nitrate in petioles, also affects the distribution of nitrate in leaves and petioles. It was found that nitrate content in petiole of *NRT1.4* mutant was significantly lower than that of wild type, but nitrate content in leaf was significantly higher than that of wild type [44].

The results showed that AsA content first increased and then decreased with the increase of nitrogen level before continuous light, but there was no significant difference. This is because the increase of nitrogen level reduced the antioxidant nutritional quality of vegetables [45,46]. Sorensen [7] found that excessive nitrogen can reduce the content of ascorbic acid in plants. Compared with that before continuous light, the content of AsA increased significantly after continuous light. As a kind of light stress, continuous light intensifies the production of reactive oxygen species (ROS), and induces the increase of antioxidant enzymes (such as catalase (CAT), superoxide dismutase (SOD), APX, DHAR, MDHAR, GR, etc.) activity and antioxidant (AsA, GSH, tocopherol, etc.) content [20,47]. AsA content and its redox state can be maintained to resist ROS production through AsA-GSH cycle [48]. Many authors have reported that the activities of enzymes (APX, MDHAR, DHAR and GR) involved in AsA regeneration in plants increase significantly under various stress conditions [48,49,50], and the overexpression of genes of these enzymes can improve the resistance of plants to various stresses [48,51]. The content of ascorbic acid depends on the rate of synthesis and decomposition and the dynamic equilibrium state of synthesis and decomposition [52].

Studies have shown that L-galactose pathway is the main biosynthetic pathway in plants, and GalLDH is the last enzyme in this pathway, which plays an important role in AsA biosynthesis [22]. The relationship between GalLDH activity and AsA content varies with plant species, environmental conditions and tissues and organs [53]. At nitrogen 12 mmol·L^−1^, the GalLDH enzyme activity of leaves under ACL (4R:1B) treatment was higher than that under ACL (2R:1B), and this changing trend was consistent with the change trend of AsA content. It showed that the higher the ratio of continuous light to red light, the higher the GalLDH enzyme activity of leaves, and the more AsA synthesized. Some studies have shown that the level of AsA content is mainly related to the level of enzyme activities such as APX, MDHAR, DHAR and GR [48,49,50]. There was no significant difference in leaf APX enzyme activity among the treatments after continuous light. It shows that APX enzyme activity has no significant effect on the increase of AsA content in this experiment. Eltayeb et al. [54]. found that the high-level expression of MDHAR in transgenic tomato improved the efficiency of transformation into AsA and reduced the conversion rate of MDHAR to DHA. At 12 mmol·L^−1^ nitrogen level, the activity of MDHAR in lettuce leaves after different continuous light quality treatments was significantly higher than that before continuous light treatment, which indicated that MDHAR enzyme promoted the increase of AsA content at 12 mmol·L^−1^ nitrogen level. After continuous light, the DHAR enzyme activity of leaves was significantly higher than that before continuous light, indicating that DHAR enzyme can promote the increase of AsA content in this experiment. Chen et al. [51] found that high-efficiency expression of DHAR genes in plants can increase the AsA content by accelerating the AsA regeneration cycle, which promotes the conversion of DHA to AsA. GSH is the electron donor of DHAR, and GSH is recovered from GSSG by GR [17]. Theoretically, the increase of GR activity is beneficial to the accumulation of AsA. In this experiment, the enzyme activity of GR decreased significantly after continuous light treatment. GR enzyme activity did not promote the increase of AsA content in leaves. Therefore, the effects of continuous light and nitrogen levels on AsA content were mainly related to GalLDH and DHAR enzyme activities. The content of AsA in leaves was much higher than that in petioles under the same treatment. It is because that the content of AsA in different tissues of the same plant is different. For example, the content of AsA in heel and other non-photosynthetic tissues is generally lower than that in photosynthetic tissues such as leaves [55]. Moreover, the activities of enzymes related to AsA biosynthesis in leaves were much higher than those in petioles, which may be one of the reasons for the low content of AsA in petioles.

## 5. Conclusions

The yield of lettuce increased slightly with the increase of nitrogen level. There was no significant difference in the soluble sugar content, nitrate content and AsA content in leaves. Compared with that before continuous light, the yield of lettuce increased significantly after continuous light. The nitrate content of lettuce decreased significantly after continuous light. High nitrogen levels (8–12 mmol·L^−1^) inhibited the accumulation of AsA in lettuce. Continuous light treatment for 72 h before harvest could promote the accumulation of AsA content in lettuce leaves under high nitrogen level. The treatment of 72 h continuous light with 4R:1B of red and blue light was beneficial to obtain higher yield and AsA content under the nitrogen level of 12 mmol·L^−1^. This is because that the combined treatment of nitrogen level and continuous light before harvest increased the GalLDH and DHAR enzyme activities involved in AsA synthesis and metabolism.

## Figures and Tables

**Figure 1 plants-10-00176-f001:**
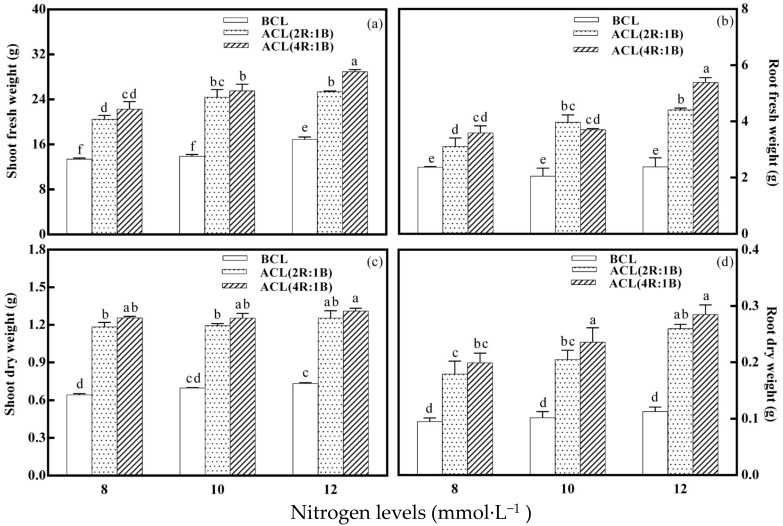
LED light quality of continuous light before harvest affects growth of hydroponic lettuce grown under increasing doses of nitrogen. Notes: BCL represents sampling before continuous light, ACL (2R:1B) represents sampling after continuous light with red and blue light quality of 2:1, ACL (4R:1B) represents sampling after continuous light with red and blue light quality of 4:1. The difference of different small letters (a, b, c, d) in the same column was statistically significant at the level of *p* < 0.05. The same below. (**a**) represents shoot fresh weight under different treatments, (**b**) represents root fresh weight under different treatments, (**c**) represents shoot dry weight under different treatments, and (**d**) represents root dry weight under different treatments.

**Figure 2 plants-10-00176-f002:**
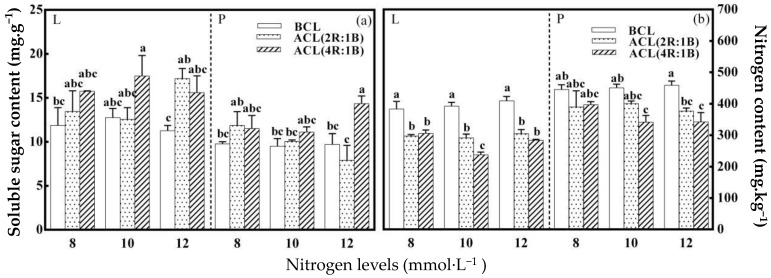
LED light quality of continuous light before harvest affects soluble sugar and nitrate content of hydroponic lettuce grown under increasing doses of nitrogen. Note: L stands for the leaves of lettuce, L stands for the leaves of lettuce. The same below. (**a**) stands for the soluble sugar content under different treatments, and (**b**) stands for nitrate content under different treatments.

**Figure 3 plants-10-00176-f003:**
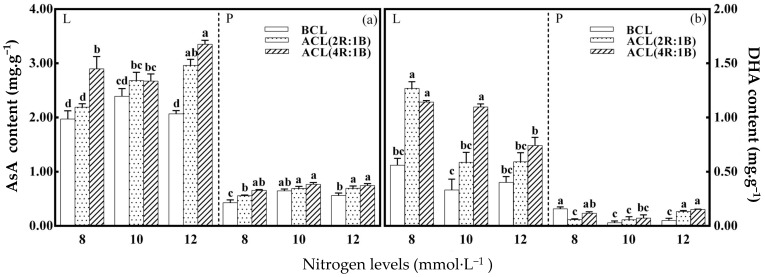
LED light quality of continuous light before harvest affects ascorbic acid (AsA) and DHA content of hydroponic lettuce grown under increasing doses of nitrogen. Note: (**a**) stands for the AsA content under different treatments, and (**b**) stands for the DHA content under different treatments.

**Figure 4 plants-10-00176-f004:**
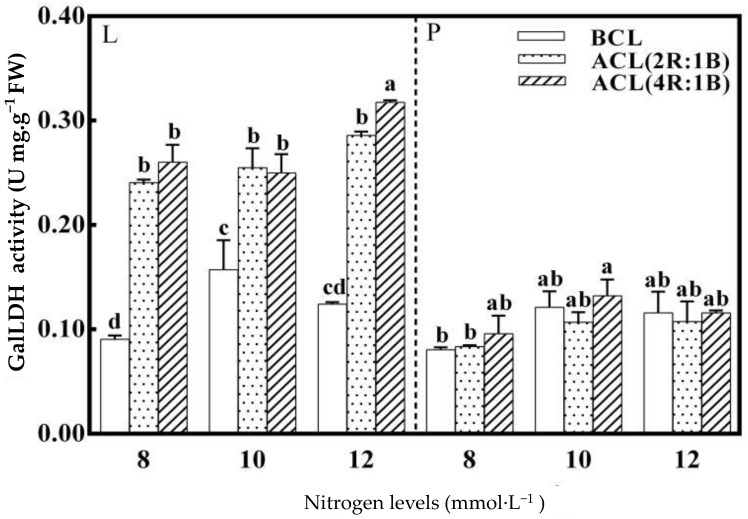
LED light quality of continuous light before harvest affects L-galactono-1,4-lactone dehydrogenase (GalLDH) enzyme of hydroponic lettuce grown under increasing doses of nitrogen.

**Figure 5 plants-10-00176-f005:**
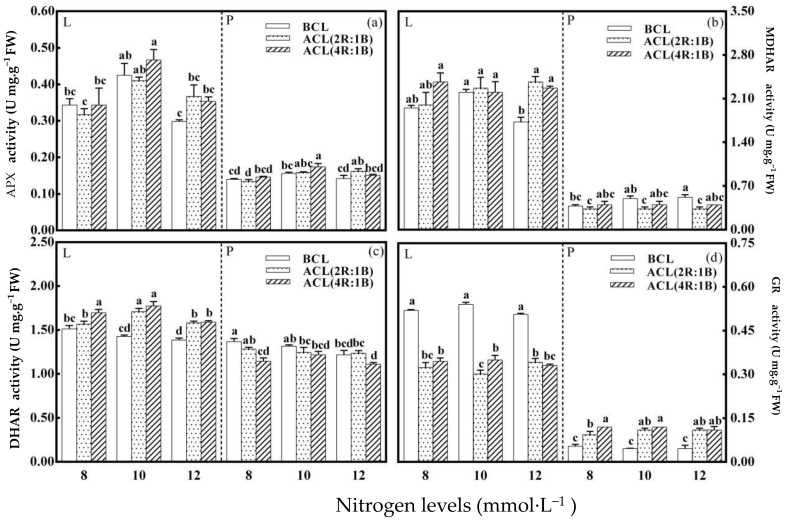
LED light quality of continuous light before harvest affects The activity of key enzymes involved in AsA recycling of hydroponic lettuce grown under increasing doses of nitrogen. Note: (**a**) stands for the APX activity under different treatments, (**b**) stands for the MDHA activity under different treatments, (**c**) stands for the DHAR activity under different treatments, and (**d**) stands for the GR activity under different treatments.

## Data Availability

Data is contained within the article.

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
