# Peer review of "LED Light Quality of Continuous Light before Harvest Affects Growth and AsA Metabolism of Hydroponic Lettuce Grown under Increasing Doses of Nitrogen"

_plants, 2021, doi:10.3390/plants10010176_

Round 1

Reviewer 1 Report

Zhang et al. presented the results of a combination of parameters (cont. light, light composition, and N levels) (mostly) on the growth and Ascorbic acid metabolism of lettuce. The overall concept is interesting, however, the presentation is rather poor and the paper feels rushed. In my opinion, 7 days typically given by Plants to revise are not sufficient, thus, the authors should take more time to improve their manuscript and re-submit for a new round of reviews. A detailed list for improvement points follow

A substantial revision concerning the use of the English language is required.

Please use as reference for C/N relations the article of Lawlor DW  Carbon and nitrogen assimilation in relation to yield: mechanisms are the key to understanding production systems.

Revise:

Nitrogen (N) is an elementis

As is known to all, leafy vegetables like to soak up nitrogen fertilizer

such as cutting off nitrogen before harvest, adding appropriate proportion of ammonium nitrogen in nutrient solution

Continuous light refers to breaking the original light-dark photoperiod change law of plants,

and providing plants with 24 h or longer light conditions. Also provide explanation for >24 light conditions, do the authors mean treatments for >24h under continuous light?

According to the current research, most studies have shown that continuous light could accelerate the growth of plants, improve the yield and quality of crops. This statement is rather misleading as many important crops develop injuries under continues light. For reference: Velez-Ramirez et al. 2014. A single locus confers tolerance to continuous light and allows substantial yield increase in tomato.

suitable for plant growth and so on

Further more, it could promote the process of nitrogen assimilation and promote the growth of plants

the short-term continuous light

L-ascorbic acid (AsA) is a kind of hexosyl lactone. Use a simpler description for AsA or provide the exact name

MDHA can produce dehydroascorbic acid (DHA), or it can be reduced to AsA again by MDHAR. 2 molecules of MDHA are oxidized/reduced spontenuesly producing a molecule of AsA and a molecule of DHA, MDHAR is reducing enzymatically MDHA to AsA using NADH as a co-factor.

DHA can be reduced to AsA or hydrolyzed to 2, 3-diketogulonic acid by dehydroascorbic acid reductase (DHAR)

Although people have realized the significance

Leafy vegetables are nitrogen-loving crops, and an effective way to increase its yield is to enhance the concentration of nitrate nitrogen in the cultivation conditions

It is hoped that this study could provide a theoretical basis for the regulation of continuous light quality in the production of high-yield and high-quality vegetables in plant factories, and provide valuable view into the interaction between continuous light quality and nutrient solution with high nitrogen level on the content and metabolism of AsA in lettuce

closed artificial light plant factory. I do not understand the terminology.

Eight lettuce plants were stochasticly selected

There were no significant differences in other yield indexes, soluble sugar content and nitrate content among different nitrogen levels. This indicated that high nitrogen level of nutrient solution treatment had inhibited the growth and nutritional quality of lettuce.too high nitrogen level would lead to the consumption of a large amount of carbohydrates in the process of nitrogen assimilation, resulting in the decrease of plant biomass under high nitrogen supply, and the growth and development of plants are inhibited. The statement that higher N inhibited the plant growth is not compatible with the results of this study, higher N did not benefit the crop but was not harmful, it is a waste of money and possibly harmful for the environment but under this regime, no differences were found. I can see that this was true for all the assays conducted by the research team. 

3.3. The Activity of Gal LDH in The Synthesis of AsA. Revise

3.4 The Activity of Key Enzymes in The Recycling of AsA. Ascorbate peroxidase is not involved in the recycling of AsA. Revise to involved in AsA metabolism.

References to figures in the discussion when results are presented can be beneficial for the reader.

The authors should discuss the redox state of AsA under the prism of the balance Photosynthesis/abiotic stress (is discussed in many papers in the literature). Their results do not suggest any photoinhibition or any other stress for lettuce, however, higher photosynthetic rates can alter AsA homeostasis. A similar reference must be added for N levels.  

Conclusions: The total yield for each treatment must be presented.

Revise/provide references

But excessive nitrogen could affect the distribution coefficient of nitrogen in Rubisco and biomechanical

components by changing the nitrogen content in plants, and then affect the photosynthetic capacity

of plant leave.

On the other hand, too high nitrogen level would lead to the consumption

Continuous light treatment before harvest can not only improve the quality of vegetables, but also increase the yield of vegetables- Also define the qualitative traits for a higher quality product.

Light quality makes a valuable contribution to plants morphogenesis, growth and development

The effect of LED light quality on AsA content and metabolism is more intricate because its spectrum can be combined in different proportions. Red light and blue light are the main light quality absorbed by crops in photosynthesis, so there are many studies on the effect of them on AsA.

High nitrogen level was not conducive to the improvement of nitrate reductase activity in lettuce leaves, and nitrate was easy to accumulate, which makes the nutritional quality of vegetables decreased. Therefore, the effect of nitrogen supply level on plant growth and development has a concentration effect, and insufficient or excessive nitrogen supply will inhibit plant growth and development.

The authors did not provide a reference or provide results concerning the Nitrate reductase activity.

And nitrogen deficiency would obstruct the synthesis of protein and decrease the activity of enzyme.

Technical.

hydroponic pots. Provide specifications for the growing medium.

The sampling before the continuous light treatment 72 h before collection was used as the control treatment. In my opinion the authors should have left a series of plants in the original light regime for control and collect it simultaneously with the treatments.

1.5% MPA. Please provide an explanation for the chemicals and list the suppliers (throughout the manuscript).

Gal LDH enzyme activity was measured with a kit. Provide the name of the kit and the manufacturer.

Other.

The figures are of low quality and must be upgraded.

Please explain the letters (a) (b) (c) and d in the figure captions.

Revise “BCL represents continuous light sampling” in the Fig1 caption

All figures must be uniform. And provide aids in the caption for a quick understanding of the figure by the reader.

In general, significant work is required in order for the figures to be in line with the requirements listed in Plants instructions for authors.

Author Response

Responses to reviewer 1

First of all, we do want to thank the reviewer for your careful and very helpful comments. We understand your comments and revise the manuscript thoroughly according to the comments. All revisions were changed into red words.

  1. Comment: A substantial revision concerning the use of the English language is required.

Answer: We had revised the abstract and whole manuscript. If necessary, we will revise it more times.

  1. Comment: Please use as reference for C/N relations the article of Lawlor DW Carbon and nitrogen assimilation in relation to yield: mechanisms are the key to understanding production systems.

Answer: Thank you for your advice. My ideas have been inspired by this article, and I quoted it. The relevant modifications are listed below.

Sufficient nitrogen could promote the photosynthesis of plants, which is closely related to the light environment of plants [7].

Lawlor, D. W. Carbon and nitrogen assimilation in relation to yield: mechanisms are the key to understanding production systems. Journal of Experimental Botany, 2002, 53(370):773-787. dol:10.1093/jexbot/53.370.773

  1. Comment: Nitrogen (N) is an elementis

As is known to all, leafy vegetables like to soak up nitrogen fertilizer

Answer: The word “elementis” was changed into “element”.

The words “soak up” was changed into “absorb”.

  1. Comment: Continuous light refers to breaking the original light-dark photoperiod change law of plants, and providing plants with 24 h or longer light conditions. Also provide explanation for >24 light conditions, do the authors mean treatments for >24h under continuous light?

Answer: Yes, 72 hours of continuous light was set before harvest in this experiment.

  1. Comment: According to the current research, most studies have shown that continuous light could accelerate the growth of plants, improve the yield and quality of crops. This statement is rather misleading as many important crops develop injuries under continues light. For reference: Velez-Ramirez et al. 2014. A single locus confers tolerance to continuous light and allows substantial yield increase in tomato.

Answer: This is my fault. The relative expression in the article is not clear. It has been explained clearly in the paper. The relevant modifications are listed below.

A large number of studies have shown that continuous light can damage the photosynthesis of eggplants and other plants[9-10]. But some studies have shown that continuous light could accelerate the growth of plants, improve the yield and quality of crops according to the current research

Murage, E.N.; Watashiro, N.; Masuda.M. Influence of light quality, PPFD and temperature on leaf chlorosis of eggplants grown under continuous illumination. Sci. Hortic. 1997, 68, 73–82,doi: 10.1016/S0304-4238(96)00953-3

  1. Comment: Further more, it could promote the process of nitrogen assimilation and promote the growth of plants

the short-term continuous light

Answer: The words “Further more” was changed into “Further-more”.

The words “the short-term” have been deleted.

  1. Comment: L-ascorbic acid (AsA) is a kind of hexosyl lactone. Use a simpler description for AsA or provide the exact name

Answer: It has been modified to “Ascorbic acid (AsA) is also known as vitamin C.”

  1. Comment: MDHA can produce dehydroascorbic acid (DHA), or it can be reduced to AsA again by MDHAR. 2 molecules of MDHA are oxidized/reduced spontenuesly producing a molecule of AsA and a molecule of DHA, MDHAR is reducing enzymatically MDHA to AsA using NADH as a co-factor.

Answer: It has been modified according to your suggestion. The relevant modifications are listed below.

2 molecules of MDHA are oxidized/reduced spontenuesly producing a molecule of AsA and a molecule of DHA, MDHAR is reducing enzymatically MDHA to AsA using NADH as a co-factor.

  1. Comment: closed artificial light plant factory. I do not understand the terminology.

Answer: It means “Plant factory with a closed environment with LED light”, and “Closed artificial light” has been deleted.

  1. Comment: Eight lettuce plants were stochasticly selected

Answer: The word “stochasticly” was changed into “randomly”.

  1. Comment: There were no significant differences in other yield indexes, soluble sugar content and nitrate content among different nitrogen levels. This indicated that high nitrogen level of nutrient solution treatment had inhibited the growth and nutritional quality of lettuce.too high nitrogen level would lead to the consumption of a large amount of carbohydrates in the process of nitrogen assimilation, resulting in the decrease of plant biomass under high nitrogen supply, and the growth and development of plants are inhibited. The statement that higher N inhibited the plant growth is not compatible with the results of this study, higher N did not benefit the crop but was not harmful, it is a waste of money and possibly harmful for the environment but under this regime, no differences were found. I can see that this was true for all the assays conducted by the research team. 

Answer: I agree with you. It has been corrected. The relevant modifications are listed below.

This showed that they have reached the asymptote, and would not continue to increase with the increase of nitrogen level.

  1. 3.4 The Activity of Key Enzymes in The Recycling of AsA. Ascorbate peroxidase is not involved in the recycling of AsA. Revise to involved in AsA metabolism.

Answer: It has been corrected. The relevant modifications are listed below.

3.4. The Activity of Key Enzyme Involved in AsA Metabolism

13.The authors should discuss the redox state of AsA under the prism of the balance Photosynthesis/abiotic stress (is discussed in many papers in the literature). Their results do not suggest any photoinhibition or any other stress for lettuce, however, higher photosynthetic rates can alter AsA homeostasis. A similar reference must be added for N levels.  

 Answer: They have been added. The relevant modifications are listed below.

Compared with that before continuous light, the content of AsA increased significantly after continuous light. The content of AsA in ACL (4R:1B) was higher than that in ACL (2R: 1B). This This may be due to increased electron transport in chloroplasts. Photosynthetic electron transport of chloroplasts is closely related to AsA pool size regulation in leaves [31-32].

Gallie D.R. The role of L-ascorbic acid recycling in responding to environmental stress and in promoting plant growth. Journal of Experimental Botany, 2012, 64(2): 433-443.dol:10.1093/jxb/ers330

Yukinori Y.;Takahiro M.;Madhusudhan R.;Ayana N.;Takashi M.;Takanori M.;Kazuya Y.;Takahiro I. and Shigeru S. Light regulation of ascorbate biosynthesis is dependent on the photosynthetic electron transport chain but independent of sugars in Arabidopsis. Journal of Experimental Botany, 2007, 58(10): 2661-2671. doi: 10.1016/j.bmcl.2009.10.110

14.Conclusions: The total yield for each treatment must be presented.

 Answer: In this paper, the measured yield of lettuce of each treatment refers to the shoot biomass average of several replicates of per plant, i.e. an average value. I think the total yield of each treatment is the average multiplied by the replicate quantity of each treatment. I think the result of comparing the total yield is the same as the average yield. the replicates.

15.Revise/provide references 

But excessive nitrogen could affect the distribution coefficient of nitrogen in Rubisco and biomechanical

components by changing the nitrogen content in plants, and then affect the photosynthetic capacity

of plant leave [2] .

Answer: They have been added.

2.Cendrero-mateo, M.P.;Carmo-silva, A.E.;Porcar-castelL, A.;Hamerlynck, E.P.;Papuga, S.A.; Moran, M.S. Dynamic response of plant chlorophyll fluorescence to light, water and nutrient availability. Functional Plant Biology, 2015, 42(8), 746-757, doi:10.1071/fp15002.

16.Continuous light treatment before harvest can not only improve the quality of vegetables, but also increase the yield of vegetables- Also define the qualitative traits for a higher quality product.

Answer: They have been added. The relevant modifications are listed below.

The production of hydroponic leafy vegetables rich in ascorbic acid (AsA) and low nitrate content has important practical significance with the improvement of people's living standards.

  1. The authors did not provide a reference or provide results concerning the Nitrate reductase activity. And nitrogen deficiency would obstruct the synthesis of protein and decrease the activity of enzyme.

Answer: It has been added.

  1. 29. Wang, X.L. The effects of nitrogen supply on photosynthetic characteristics in leaves of tobacco seedlings.CAAS,2019.05 (In Chinese)

18.Technical. hydroponic pots. Provide specifications for the growing medium.

 Answer: The specifications of hydroponic pots is length×width×height=180 × 60 × 6 cm.

19.The sampling before the continuous light treatment 72 h before collection was used as the control treatment. In my opinion the authors should have left a series of plants in the original light regime for control and collect it simultaneously with the treatments.

Answer: This is also an experimental idea, which I used in previous experiments. (You can see it in this paper. Improvement effects of Red and Blue LED continuous lighting before harvest on quality of hydroponic lettuce,doi:10.3969/j.issn.1000-6362.2020.07.004)However, I think it is reasonable to use the sample at the beginning of continuous light before harvest for comparison.

20.1.5% MPA. Please provide an explanation for the chemicals and list the suppliers (throughout the manuscript).

Gal LDH enzyme activity was measured with a kit. Provide the name of the kit and the manufacturer.

Answer: 1.5% Metaphosphoric acid (Sigma-aldrich, USA)

Gal LDH enzyme activity was measured with a kit (Solarbio, China).

And all related context has been revised.

21.The figures are of low quality and must be upgraded. Please explain the letters (a) (b) (c) and d in the figure captions.Revise “BCL represents continuous light sampling” in the Fig1 caption All figures must be uniform. And provide aids in the caption for a quick understanding of the figure by the reader.

Answer: They have been added.

Notes: BCL represents sampling before continuous light, ACL (2R:1B) represents continuous light processing of pre mining led with red blue light quality ratio of 2:1, ACL (4R:1B) represents continuous light processing of pre mining led with red blue light quality ratio of 4:1, (a) represents shoot fresh weight under different treatments, (b) represents root fresh weight under different treatments, (c) represents shoot dry weight under different treatments,(d) represents root dry weight under different treatments. The same below.

And all related context has been revised.

Reviewer 2 Report

The research is original and provides an advance in the current knowledge on the use of led light to affect grown, quality, and metabolism of plants in soilless cultivation. The methods are described with details (but some clarification is needed) and the results are appropriately interpreted and significant: a continuous illumination with led light before harvest can promote a decrease of nitrate content and an increase of ascorbic acid in leaves. The article is nice but the feeling is that it was written and sent too quickly and without proper control, so a minor revision is needed.

Specific comments: review the spacing before and after the periods and commas throughout the text.

Introduction:

line 35:   element or component of protein

line 39: Rubisco to define and specify for the first time (not everyone remembers or knows)

line 43: inhibited

line 45: AsA to define and specify for the first time

line 60: to link 'further more'

line 73: there is report.....maybe there is a report or it is reported.....

line 85: catalyzed

line 94: its - their yield

Materials and methods

110: you should indicate how many plants per m2

117: are the 8 plants sampled for the initial value the control treatment? in line 205 is BCL the control treatment?

117: continuous light

118-119: are Q2 and ACL (2R:1B) the same?  Q4 or ACL (4R:1B)? What does ACL stand for?

122: are 655etc... were

128: what did you use for reaching this pH? if you add a different amount of KNO3 and KCL you should indicate a range of EC.

Some doubts:

  • Sonneveld C. suggests for lettuce 19 mmol / l of NO3 for the standard nutrient solution in circulating water .. So, is it more right to tell about “high values” (8-10-12 mmol / l NO3) or is it more correct tell about “increasing doses of nitrogen”? Moreover in the title you write "different nitrogen levels"....
  • if I have understood ....the plants were transplanted after the second true leaf, they were exposed to red and blue light (4:1) for 17 days, 8 of them were used as the control, for the others, the light quality was maintained (4:1) and changed (2:1) for 72 h and after they were harvested. So, were the lettuce plants harvested after about one month? Why hasn't the maturity for commercial sale been reached?

132: I confess I do not understand what "stochasticly" means in this context

136:137: "The leaves and petioles frozen with liquid nitrogen" are repeated

138: .last. ?? maybe a typo....

159: Then lowercase

160:  10 µl of reducing solution is added...was added. Delete the repetition of  '10 µl of reducing solution' at the end

161: add....50 µl of reaction stop solution was added and shaken well.

174: AO solution - to define and specify

187: Shoot lowercase

192: SDM or SDW? check and standardize even the others

194: 4R:1B ....ACL (4R:1B)

205-206:  typo: ACL(2R:1R) …..4R:1R

214:  "in the perspective of the entire growth period" is not clear. what do you mean?

233- 250: you must indicate Fig. 3 and Fig. 4

241: 2-4 b.....maybe a typo....

245-287: ACL (2R:1B)

252: When lowercase

Discussion

297: enzyme or enzymes? if it is one, which?

302: concentration effect  - to clarify

307: Too - uppercase

310: it is a repetition of 296-297

339: maybe "changing trend"

Conclusion

361:  There was no soluble sugar content, nitrate content and AsA content in leaves....to check.

In the conclusions, you should mention the fate of nitrates as in lines 320-322

Author Response

Responses to reviewer 2

First of all, we do want to thank the reviewer for your careful and very helpful comments. We understand your comments and revise the manuscript thoroughly. All revisions were changed into red words.

  1. Comment: Specific comments: review the spacing before and after the periods and commas throughout the text.

Answer: We had revised the spacing before and after the periods and commas throughout the text.

  1. Comment: line 35:   element or component of protein

Answer: It has been modified according to your suggestion.

  1. Comment: Rubisco to define and specify for the first time (not everyone remembers or knows)

Answer: The full name has been added. “Ribulose-1,5-bisphosphate carboxylase/oxygenase (Rubisco)”

  1. line 45: AsA to define and specify for the first time

Answer: “ascorbic acid (AsA)” has been added

5.line 60: to link 'further more'

Answer: The words “further more” was revised into “further-more”.

  1. line 73: there is report.....maybe there is a report or it is reported.....

Answer: It has been revised. It was revised into “ There is a report that…”

  1. line 85: catalyzed

Answer: The word “catalysed” was revised into “catalyzed”.

  1. line 94: its - their yield

Answer: The word “its” was revised into “their”.

Materials and methods

9.110: you should indicate how many plants per m2

Answer: The sentence has been added.

In terms of planting density, there are about 36 lettuce plants per square meter.

10.117: are the 8 plants sampled for the initial value the control treatment? in line 205 is BCL the control treatment?

Answer: Sorry, I didn't make it clear. This is not a control treatment, but a comparison of two samples in time sequence.

  1. 118-119: are Q2 and ACL (2R:1B) the same?  Q4 or ACL (4R:1B)? What does ACL stand for?

Answer: ACL (2R:1B) represents continuous light processing of pre mining led with red blue light quality ratio of 2:1, ACL (4R:1B) represents continuous light processing of pre mining led with red blue light quality ratio of 4:1. They have been unified as ACL (2R:1B) and ACL (4R:1B).

12.122: are 655etc... were

Answer: The word “are” was revised into “were”.

  1. 128: what did you use for reaching this pH? if you add a different amount of KNO3 and KCL you should indicate a range of EC.

Answer: The prepared nutrient solution showed acidity, because there are a lot of NO3-1 and SO4-2. It achieves the required pH by adding KOH .EC: 1.3,1.5,1.7 mS·cm-1

  1. Some doubts:
  • Sonneveld C. suggests for lettuce 19 mmol / l of NO3 for the standard nutrient solution in circulating water .. So, is it more right to tell about “high values” (8-10-12 mmol / l NO3) or is it more correct tell about “increasing doses of nitrogen”? Moreover in the title you write "different nitrogen levels"....

Answer: Firstly, the treatment of 19mm nitrogen level and the setting of nitrogen level in this experiment are both higher nitrogen level settings. Nitrogen is easily absorbed by leafy vegetables, which will be stored in vacuoles after absorption, resulting in a large amount of nitrate accumulation. I think you have a better point of view on the topic, so I changed the title. “LED Light quality of continuous Light before Harvest Affects Growth and AsA Metabolism of Hydroponic Lettuce Grown under Increasing Doses of Nitrogen”

  • if I have understood ....the plants were transplanted after the second true leaf, they were exposed to red and blue light (4:1) for 17 days, 8 of them were used as the control, for the others, the light quality was maintained (4:1) and changed (2:1) for 72 h and after they were harvested. So, were the lettuce plants harvested after about one month? Why hasn't the maturity for commercial sale been reached?

Answer: Because the variety of lettuce is different, the growth characteristics are different, and the harvest time is also different. The size of this variety at this time is the best harvest time and has better taste.

  1. 132: I confess I do not understand what "stochasticly" means in this context

Answer: The word “stochasticly” was revised into “randomly”.

16.136:137: "The leaves and petioles frozen with liquid nitrogen" are repeated

Answer: It has been deleted.

17.138: .last. ?? maybe a typo....

Answer: It has been deleted.

18.159: Then lowercase

Answer: The word “Then” was revised into “then”.

19.160:  10 µl of reducing solution is added...was added. Delete the repetition of  '10 µl of reducing solution' at the end

Answer: The sentence has been revised into” 10 µl of reducing solution was added and incubated at 25 ℃for 45 min.”

20.161: add....50 µl of reaction stop solution was added and shaken well. Answer: The sentence has been revised into” 50 µl of reaction stop solution was added and shake well

21.174: AO solution - to define and specify

Answer: The word “AO” was revised into “Ascorbic acid oxidase”.

22.187: Shoot lowercase

Answer: The word “Shoot” was revised into “shoot”.

23.192: SDM or SDW? check and standardize even the others

Answer: It has been revised into” SDW”.

24.194: 4R:1B ....ACL (4R:1B)

Answer: The words “4R:1B” was revised into “ACL (4R:1B)”.

25.205-206:  typo: ACL(2R:1R) …..4R:1R

Answer: They have been revised. The relevant modifications are listed below.

Notes: BCL represents sampling before continuous light, ACL (2R:1B) represents continuous light processing of pre mining led with red blue light quality ratio of 2:1, ACL (4R:1B) represents continuous light processing of pre mining led with red blue light quality ratio of 4:1, (a) represents shoot fresh weight under different treatments, (b) represents root fresh weight under different treatments, (c) represents shoot dry weight

ACL (2R:1B) represents continuous light processing of pre mining led with red blue light quality ratio of 2:1, ACL (4R:1B) represents continuous light processing of pre mining led with red blue light quality ratio of 4:1

  1. 214:  "in the perspective of the entire growth period" is not clear. what do you mean?

Answer: I think it is unnecessary. It has been deleted.

27.233- 250: you must indicate Fig. 3 and Fig. 4

Answer: It has been revised. The relevant modifications are listed below.

Note: (a) represents the AsA content under different treatments, (b) represents the DHA content under different treatments.

28.241: 2-4 b.....maybe a typo....

Answer: It has been deleted.

29.245-287: ACL (2R:1B)

Answer: The words “2R:1B” was revised into “ACL (2R:1B)”.

30.252: When lowercase

Answer: The words “When” was revised into “when”.

Discussion

31.297: enzyme or enzymes? if it is one, which?

Answer: It has been revised into” enzyme

32.302: concentration effect  - to clarify

Answer: Therefore, the effect of nitrogen supply level on plant growth and development has a concentration effect, and insufficient or excessive nitrogen supply will limit plant growth and development.

33.339: maybe "changing trend"

Answer: It has been revised into” changing trend

34.Conclusion

361:  There was no soluble sugar content, nitrate content and AsA content in leaves....to check.

In the conclusions, you should mention the fate of nitrates as in lines 320-322

Answer: It has been revised. The relevant modifications are listed below.

There was no significant difference in the soluble sugar content, nitrate content and AsA content in leaves.

Compared with that before continuous light, the yield of lettuce increased significantly after continuous light.And the nitrate content of lettuce decreased significantly after continuous light.

Round 2

Reviewer 1 Report

The authors of the manuscript addressed some of the points for improvement proposed in my previous review, however, I feel that the manuscript is still unsuitable for publication mainly because of the incomplete discussion (some parts belong in the results part, while the authors fail to describe the phenomenon and provide a useful physiological explanation) and because of the poor use of the English language (I strongly suggest that the authors should use the experience of a professional proof reader as no many typos exist (some do exist), however, the choice of sentences is rather poor).  Some points for improvment from my previous review were not addressed for example the reference of Nitrate reductase in line 340 (no citation of results are presentes), the quality of figures and legends requires significant imporvent.

In conclusion, the manuscript is still incomplete and unfit for publication in it current state. I urge the authors to take their time an work towards the improvment of the presentation of their results and provide useful conclusions (especially explaining the physiological phenomenon) drown by their findings. 

Author Response

First of all, we do want to thank the reviewer for your careful and very helpful comments. We understand your comments and revised the manuscript thoroughly. All revisions in the manuscript were changed into blue words.

Comments: The authors of the manuscript addressed some of the points for improvement proposed in my previous review, however, I feel that the manuscript is still unsuitable for publication mainly because of the incomplete discussion (some parts belong in the results part, while the authors fail to describe the phenomenon and provide a useful physiological explanation) and because of the poor use of the English language (I strongly suggest that the authors should use the experience of a professional proof reader as no many typos exist (some do exist), however, the choice of sentences is rather poor).  Some points for improvement from my previous review were not addressed for example the reference of Nitrate reductase in line 340 (no citation of results are presents), the quality of figures and legends requires significant improvement.

Answer: 1. We have carefully revised the discussion part by adding several references, and the language was improved greatly. All revisions in the manuscript were changed into blue words.

  1. The title, tick marks, units, and notes of the figures' axis have been modified.
  2. In addition, several references were cited and added.

Added References

  1. Parisi, M;Giordano, L;Pentangelo,A;D'Onofrio,B;Villari G. Effects of different levels of nitrogen fertilization on yield and fruit quality in processing tomato. In International Symposium Towards Ecologically Sound Fertilisation Strategies for Field Vegetable Production.2004,700: 129-132.
  2. Jacques, L. B.;Camille, B.;Christophe, R.;Frédéric, B.;Stéphane, A. The 'trade-off' between synthesis of primary and secondary compounds in young tomato leaves is altered by nitrate nutrition: experimental evidence and model consistency. Journal of Experimental Botany, 2009, 60(15): 4301-4314.doi: 10.1093/jxb/erp271
  3. Bian, Z.H.; Yang, Q.C.;Liu,W.K. Effects of light quality on the accumulation of phytochemicals in vegetables produced in controlled environments: A review. J. Sci. Food Agr.2015. 95:869–877.doi:10.1002/jsfa.6789
  4. Zha, L.Y.;Liu, W.K.;Yang, Q.C.;Zhang, Y.B.;Shao, M.J. Regulation of Ascorbate Accumulation and Metabolism in Lettuce by the Red:Blue Ratio of Continuous Light Using LEDs. Frontiers in Plant Science, 2020, 11.doi: 10.3389/fpls.2020.00704
  5. Bian, Z.H.; Yang, Q.C.;Li, T.;Cheng, R.F.;Barnett, Y. Study of the beneficial effects of green light on lettuce grown under short-term continuous red and blue light-emitting diodes. Physiologia Plantarum, 2018, 164: 226-240.doi: 10.1111/ppl.12713
  6. Bian, Z.H.;Cheng, R.F.; Yang, Q.C.;Wang, J. Continuous light from red, blue, and green light-emitting diodes reduces nitrate content and enhances phytochemical concentrations and antioxidant capacity in lettuce.J. Amer. Soc. Hort. Sci. 2016, 141(2):186–195.doi: 10.21273/JASHS.141.2.186
  7. Martinoia, E.;Heck, U.;Wiemken, A. Vacuoles as storage compartments for nitrate in barley leaves. Nature, 1981, 289: 292-294. doi: 10.1038/289292a0
  8. Wang, Y.Y.;Hsu, P.K.;Tsay, Y.F. Uptake, allocation and signaling of nitrate. Trends in plant science, 2012, 17: 458-467. doi: 10.1016/j.tplants.2012.04.006
  9. Aires, A.;Rosa, E.;Carvalho, R. Effect of nitrogen and sulfur fertilization on glucosinolates in the leaves and roots of broccoli sprouts (Brassica oleracea var italica). Journal of the Science of Food and agriculture. 2006,86: 1512-1516. doi: 10.1002/jsfa.2535
  10. Liu, D.;Liu, W.;Zhu, D.;Geng, M.;Zhou, W.;Yang, T. Nitrogen effects on total flavonoids, chlorogenic acid, and antioxidant activity of the medicinal plant Chrysanthemum morifolium. Journal of Plant Nutrition and Soil Science. 2010, 173,268-274. doi:10.1002/jpln.200900229
  11. Sorensen, J.N. Use of the Nmin-method for optimization of vegetable nitrogen nutrition. Acta Hotr, 1993, 339: 79-192
  12. Ntagkas, N., Woltering, E. J., and Marcelis, L. F. M. Light regulates ascorbate in plants: an integrated view on physiology and biochemistry. Environ. Exp. Bot.2018,147, 271–280. doi: 10.1016/j.envexpbot.2017.10.009 
  13. Smirnoff N. Vitamin C: biosynthesis of vitamins in plants. Advances in Botanical Research, 2011, 59, 107-177. doi: 10.1016/b978-0-12-385853.00003-9.

Round 3

Reviewer 1 Report

After another careful reading of the manuscript, I have found that the quality of the manuscript requires further improvment. I urge the authors to consult with the author services of MDPI to improve the presentation. I have made some specific recommendations for improvement below. 

Line 27 change “Gal LDH and DHAR involved in AsA synthesis” to “GalLDH and DHAR involved in AsA synthesis and in the recycling of DHAR to AsA respectively”

Line 31 change “element or component of protein…” to “ element which is a component of proteins….”

Line 31 change “As is known to all, leafy vegetables like to absorb nitrogen fertilizer…” to

“ The production of leafy vegetables requires high doses of nitrogen fertilizer…”

Line 35 “excessive nitrogen could affect the distribution coefficient” what is “distribution coefficient of RuBiSCo? Preferably change to “can alter the plant physiology primarily the photosynthetic capacity of plants.”

Line 37 change “On the other hand, too high nitrogen level would lead to the consumption of a large amount of carbohydrates…” to “The application of extensive N fertilizers, however, may lead to the consumption of a large amount of carbohydrates…”

Line 41 provide reference

Line 42 “Previous studies have shown that there are some kinds of methods to enhance the quality of hydroponic leafy vegetables” change to “Previous studies have shown that the enhancement of the hydroponic leafy vegetables quality can be achieved with the employment of methods such as…”

Line 47 “…important practical significance with the improvement of people's living standards…” change to “is critical for the improvement of the cultivation systems towards the production of high-quality products for the consumers”

Line 50 “that nitrogen nutrition not make an appreciable difference [4] or increase it [5], but

excessive nitrogen will reduce the ascorbic acid content in plants [6]” change to “contradictive results exist on the effect of the levels of N fertilization on the AsA levels [4,5]. Moreover, higher than recommended doses of N can even result in the reduction of the levels of the antioxidant [6]. ”

Line 55 Continuous light refers to breaking the original light-dark photoperiod change law of plants…. What is light-dark photoperiod change law? Rephrase.

Line 57 “A large number of studies have shown that continuous light can damage the photosynthesis of eggplants and other plants” rephrase to “A large number of studies have shown that continuous light can damage the photosynthesis many plant species”

Line 58 “But some studies have shown that continuous light could accelerate the

growth of plants, improve the yield and quality of crops according to the current research” rephrase to “A growing body of evidence, however, shows that continuous light could accelerate the growth, improve the yield and quality of specific crops” and continue by merging the following sentence revising it to “this can be achieved by the improvement of the photosynthetic capacity via the prolonging the time of light exposure, strengthening the light intensity and by selecting the proportion of light quality that stimulates plant growth”

Line 90 provide reference

Line 104-108 remove this sentence

Line 319 revise

Line 320 “This showed that they have reached the asymptote, and would not continue to increase with the increase of nitrogen level” revise to “This showed that the biomass accumulation has reached the asymptote, and would not continue to increase with the increase of nitrogen level”

Line 322 “Therefore, how to help hydroponic plants make full use of nitrogen in nutrient solution environment is particularly important” revise to “Therefore, the optimization of nitrogen assimilation of hydroponically cultivated plants is particularly important”

Line 326 provide specific examples that continuous light is beneficial to specific crops.

Line 331 put specific references to tables and figures when discussing the results of this study, do so for every such reference throughout the manuscript.

Line 331 “This is accordant” revise to “this is according” plus provide more references (more than the one that is provided”

Line 339 provide reference

Lines 347-350 I do not understand this sentence are “proteases” “proteins”? “and chlorophyll in the process of photosynthesis” what is the meaning? Do the authors mean that proteins such as RuBiSCo are involved in the biomass accumulation via the process of photosynthesis, thus, the improved N utilization can be promotive for plant productivity?

Line 358 “accommodate” is accumulate?

Line 374 please refer to the special issue of fronters in plant science under the title “The Role of Light in Abiotic Stress Acclimation” to find a more suitable physiological explanation. I believe that AsA is accumulated when light intensity is stressful for plants.

Line 406 different tissue composition of each plant organ leads to different concentrations of AsA, this is not necessarily a result of the enzyme activity of the AsA biosynthesis/accumulation enzymes.

Author Response

Responses to Reviewer 1 (Round 3) Comments

Thank you very much for your advice. We understand your comments and revised the manuscript thoroughly. All revisions were changed into green words.

1.Comments: After another careful reading of the manuscript, I have found that the quality of the manuscript requires further improvement. I urge the authors to consult with the author services of MDPI to improve the presentation. I have made some specific recommendations for improvement below. 

 Answer: We have revised and improved the manuscript greatly. If necessary, we will revise it more times.

  1. Comments: Line 27 change “Gal LDH and DHAR involved in AsA synthesis” to “GalLDH and DHAR involved in AsA synthesis and in the recycling of DHAR to AsA respectively”

Answer: The words “Gal LDH and DHAR involved in AsA synthesis” were revised into “GalLDH and DHAR involved in AsA synthesis and in the recycling of DHAR to AsA respectively”.

  1. Comments: Line 31 change “element or component of protein…” to “ element which is a component of proteins….”

Answer: The words “element or component of protein…” were revised into “element which is a component of proteins….”.

  1. Comments: Line 31 change “As is known to all, leafy vegetables like to absorb nitrogen fertilizer…” to “The production of leafy vegetables requires high doses of nitrogen fertilizer…”

Answer: The words “As is known to all, leafy vegetables like to absorb nitrogen fertilizer…” were revised into “The production of leafy vegetables requires high doses of nitrogen fertilizer”.

5.Comments: Line 35 “excessive nitrogen could affect the distribution coefficient” what is “distribution coefficient of RuBiSCo? Preferably change to “can alter the plant physiology primarily the photosynthetic capacity of plants.”

Answer: The words “affect the distribution coefficient” were revised into “alter the plant physiology primarily the photosynthetic capacity of plants.”.

6.Comments: Line 37 change “On the other hand, too high nitrogen level would lead to the consumption of a large amount of carbohydrates…” to “The application of extensive N fertilizers, however, may lead to the consumption of a large amount of carbohydrates…”

Answer: It refers to the distribution of nitrogen content in different physiological activities such as Rubisco and biomechanical components.

7.Comments: Line 41 provide reference

Answer: Its reference has been added.

  1. Zhou,W.L.;Liu, W.K.;Yang, Q.C. Reducing nitrate content in lettuce by pre-harvest continuous light delivered by red and blue light-emitting diodes.Journal of Plant Nutrition, 2013, 36(3):481-490. doi: 10.1080/01904167.2012.748069

8.Comments: Line 42 “Previous studies have shown that there are some kinds of methods to enhance the quality of hydroponic leafy vegetables” change to “Previous studies have shown that the enhancement of the hydroponic leafy vegetables quality can be achieved with the employment of methods such as…”

Answer: The words “Previous studies have shown that there are some kinds of methods to enhance the quality of hydroponic leafy vegetables” were revised into “Previous studies have shown that the enhancement of the hydroponic leafy vegetables quality can be achieved with the employment of methods such as...”.

9.Comments: Line 47 “…important practical significance with the improvement of people's living standards…” change to “is critical for the improvement of the cultivation systems towards the production of high-quality products for the consumers”

Answer: The words “…important practical significance with the improvement of people's living standards…” were revised into “is critical for the improvement of the cultivation systems towards the production of high-quality products for the consumers”.

10.Comments: Line 50 “that nitrogen nutrition not make an appreciable difference [4] or increase it [5], but excessive nitrogen will reduce the ascorbic acid content in plants [6]” change to “contradictive results exist on the effect of the levels of N fertilization on the AsA levels [4,5]. Moreover, higher than recommended doses of N can even result in the reduction of the levels of the antioxidant [6]. ”

Answer: The words “…important practical significance with the improvement of people's living standards…” were revised into “contradictive results exist on the effect of the levels of N fertilization on the AsA levels [5,6]. Moreover, higher than recommended doses of N can even result in the reduction of the levels of the antioxidant [7]”.

11.Comments: Line 55 Continuous light refers to breaking the original light-dark photoperiod change law of plants…. What is light-dark photoperiod change law? Rephrase.

Answer: The sentence was revised into “Continuous light refers to breaking the original light-dark cycle change in 24 hours of plants….

Continuous light is a special light mode that can maximize the lighting time, and providing plants with 24 h or over a period of more than 24 hours in most cases.

12.Comments: Line 57 “A large number of studies have shown that continuous light can damage the photosynthesis of eggplants and other plants” rephrase to “A large number of studies have shown that continuous light can damage the photosynthesis many plant species”

Answer: The words “A large number of studies have shown that continuous light can damage the photosynthesis of eggplants and other plants” were revised into “A large number of studies have shown that continuous light can damage the photosynthesis many plant species”.

13.Comments: Line 58 “But some studies have shown that continuous light could accelerate the growth of plants, improve the yield and quality of crops according to the current research” rephrase to “A growing body of evidence, however, shows that continuous light could accelerate the growth, improve the yield and quality of specific crops” and continue by merging the following sentence revising it to “this can be achieved by the improvement of the photosynthetic capacity via the prolonging the time of light exposure, strengthening the light intensity and by selecting the proportion of light quality that stimulates plant growth”

Answer: They have been revised. The revised sentences are as follows.

A growing body of evidence, however, shows that continuous light could accelerate the growth, improve the yield and quality of specific crops [12]. This can be achieved by the improvement of the photosynthetic capacity via the prolonging the time of light exposure, strengthening the light intensity and by selecting the proportion of light quality that stimulates plant growth.

14.Comments: Line 90 provide reference

Answer: Their references have been added.

  1. Zha, L.Y.;Zhang, Y.B.;Liu, W.K. Dynamic responses of ascorbate pool and metabolis m in lettuce to long-term continuous light provided by red and blue LEDs. Environmental and Experimental Botany, 2019, 7(163): 15-23. doi:10.1016/j.envexpbot.2019.04.003.
  2. Zha, L.Y.;Liu, W.K.;Zhang, Y.B.;Shao, M.J. Morphological and Physiological Stress Responses of Lettuce to Different Intensities of Continuous Light. Frontiers in plant science, 2019, 10:1440-1440. doi:10.3389/fpls.2019.01440
  3. Comments: Line 104-108 remove this sentence

 Answer: They have been removed.

  1. Comments: Line 319 revise

Answer: It has been revised.

There were no significant differences in other yield indexes among different nitrogen levels.

17.Comments: Line 320 “This showed that they have reached the asymptote, and would not continue to increase with the increase of nitrogen level” revise to “This showed that the biomass accumulation has reached the asymptote, and would not continue to increase with the increase of nitrogen level”

Answer: The words “This showed that they have reached the asymptote, and would not continue to increase with the increase of nitrogen level” were revised into “This showed that the biomass accumulation has reached the asymptote, and would not continue to increase with the increase of nitrogen level”.

18.Comments: Line 322 “Therefore, how to help hydroponic plants make full use of nitrogen in nutrient solution environment is particularly important” revise to “Therefore, the optimization of nitrogen assimilation of hydroponically cultivated plants is particularly important”

Answer: The words “Therefore, how to help hydroponic plants make full use of nitrogen in nutrient solution environment is particularly important” were revised into “Therefore, the optimization of nitrogen assimilation of hydroponically cultivated plants is particularly important”.

19.Comments: Line 326 provide specific examples that continuous light is beneficial to specific crops.

Answer:Continuous light could promote the yield of sweet pepper [37], the growth of young tea plants [38] and the germination rate of alfalfa seeds [9]. Therefore, it is particularly important to explore the effect of continuous light quality on the growth of leafy vegetables under different gradient nitrogen levels.

20.Comments: Line 331 put specific references to tables and figures when discussing the results of this study, do so for every such reference throughout the manuscript.

Line 331 “This is accordant” revise to “this is according” plus provide more references (more than the one that is provided”

Answer: It has been modified on the basis of your suggestion.The revised sentence is as follows.

This is according with the former research results [3,39,40], which may be because the increase of photosynthetic duration to improve the net photosynthetic efficiency of lettuce leaves and promote the accumulation of photosynthetic products, thus significantly improving the yield of lettuce [9].

21.Comments: Line 339 provide reference

Answer:It has been modified on the basis of your suggestion.The revised sentence is as follows.

It is because red light is the most absorbed light by green plants [40].

  1. Liu, W. K.;Yang, Q. C. Semiconductor lighting for protected horticulture production.Press: China Agricultural Science and Technology Press,2016.(In Chinese)

22.Comments: Lines 347-350 I do not understand this sentence are “proteases” “proteins”? “and chlorophyll in the process of photosynthesis” what is the meaning? Do the authors mean that proteins such as RuBiSCo are involved in the biomass accumulation via the process of photosynthesis, thus, the improved N utilization can be promotive for plant productivity?

Answer: It is that our expression is not rigorous. We have made the following changes in the article. 

Nitrogen is a component of photosynthetic proteins and pigments, so the content and activity of photosynthetic proteins and pigments can be changed by altering the nitrogen level, which affects the photosynthesis and biomass of plants.

23.Comments: Line 358 “accommodate” is accumulate?

Answer: The word “accommodate” was revised into “accumulation”.

24.Comments: Line 374 please refer to the special issue of frontiers in plant science under the title “The Role of Light in Abiotic Stress Acclimation” to find a more suitable physiological explanation. I believe that AsA is accumulated when light intensity is stressful for plants.

Answer: They have been revised. The revised sentences are as follows.

As a kind of light stress, continuous light intensifies the production of reactive oxygen species (ROS), and induces the increase of antioxidant enzymes (such as catalase (CAT), superoxide dismutase (SOD), APX, DHAR, MDHAR, GR, etc.) activity and antioxidant (AsA, GSH, tocopherol, etc.) content [21,49]. AsA content and its redox state can be maintained to resist ROS production through AsA-GSH cycle [50]. Many authors have reported that the activities of enzymes (APX, MDHAR, DHAR and GR) involved in AsA regeneration in plants increase significantly under various stress conditions [50-52], and the overexpression of genes of these enzymes can improve the resistance of plants to various stresses [50,53].

25.Comments: Line 406 different tissue composition of each plant organ leads to different concentrations of AsA, this is not necessarily a result of the enzyme activity of the AsA biosynthesis/accumulation enzymes. 

Answer: It is that my expression is not rigorous. This is the reasoning based on the experimental data. I have added "may be" to the sentence.The revised sentence is as follows.

Moreover, the activities of enzymes related to AsA biosynthesis in leaves were much higher than those in petioles, which may be one of the reasons for the low content of AsA in petioles.
